# Gaussian Process Bandit Optimization of the Thermodynamic Variational Objective

**Vu Nguyen**
University of Oxford
vu@robots.ox.ac.uk

**Vaden Masrani**
University of British Columbia
vadmas@cs.ubc.ca

**Rob Brekelmans**
USC Information Sciences Institute
brekelma@usc.edu

**Michael A. Osborne**
University of Oxford
mosb@robots.ox.ac.uk

**Frank Wood**
University of British Columbia
fwood@cs.ubc.ca

## Abstract

Achieving the full promise of the Thermodynamic Variational Objective (TVO), a recently proposed variational lower bound on the log evidence involving a one-dimensional Riemann integral approximation, requires choosing a "schedule" of sorted discretization points. This paper introduces a bespoke Gaussian process bandit optimization method for automatically choosing these points. Our approach not only automates their one-time selection, but also dynamically adapts their positions over the course of optimization, leading to improved model learning and inference. We provide theoretical guarantees that our bandit optimization converges to the regret-minimizing choice of integration points. Empirical validation of our algorithm is provided in terms of improved learning and inference in Variational Autoencoders and Sigmoid Belief Networks.

## 1 Introduction

The Variational Autoencoder (VAE) framework has formed the basis for a number of recent advances in unsupervised representation learning [17, 35, 41]. Assuming a generative model involving latent variables, VAEs perform maximum likelihood parameter estimation by optimizing the tractable Evidence Lower Bound (ELBO) on the logarithm of the model evidence. In doing so, the VAE framework introduces an inference network, which seeks to approximate the true posterior over latent variables. While the ELBO is a common choice of variational inference objective, recent work has sought to improve the model learning [7, 38, 30, 25] or inference aspects [34, 18, 8, 12] of this task.

In this work, we build upon the recent Thermodynamic Variational Objective (TVO), which frames log-likelihood estimation as a one-dimensional integral over the unit interval [26]. The integral is estimated using a Riemann sum approximation, as visualized in Figure 1, yielding a natural family of variational inference objectives which generalize and tighten the ELBO.

The choice of a $d$-dimensional vector of points $\boldsymbol{\beta} = [\beta_0, \beta_1, ..., \beta_{d-1}]^T$ at which to construct this numerical approximation is an important hyperparameter for the TVO, which we refer to as an "integration schedule" throughout this work. Previous work [26] uses a static integration schedule, and requires grid search over the choice of initial $\beta_1$. However, since the shape of the integrand reflects the quality of the inference network (§2), recent work [6] suggests that this scheduling procedure may be improved by dynamically choosing $\boldsymbol{\beta}$ over the course of training. Our proposed approach also allows the TVO to be adapted to different model architectures and schedule dimensionality without the need for grid search.

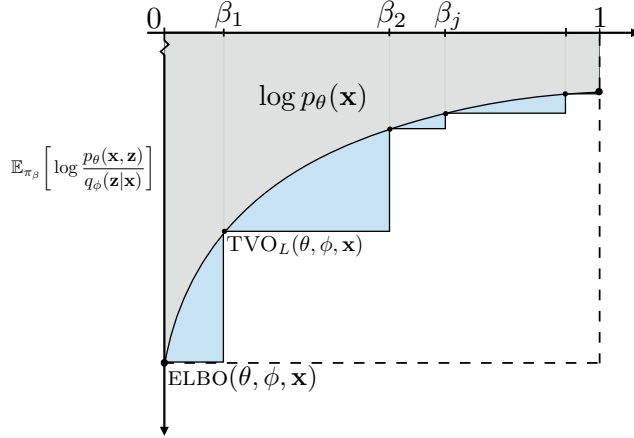

Figure 1: The TVO objective frames log likelihood estimation as a Riemann sum approximation to a 1-d integral, with the ELBO as a special case for a single $\beta_0 = 0$. The TVO (area in blue) bounds the integral more tightly than the ELBO (area within dotted lines).

Our primary contribution is to automate the choice of integration schedules using a Gaussian process bandit optimization. We first demonstrate that maximizing the TVO objective as a function of $\boldsymbol{\beta}$ is equivalent to a regret-minimization problem, where the black-box reward function reflects improvement in the objective for a given choice of schedule. We model this reward function over the course of training epochs using a time-varying Gaussian process (GP). Our entire procedure amounts to 1) choosing $\boldsymbol{\beta}$ to maximize an acquisition function in our surrogate GP model, 2) observing the reward function as the improvement in the TVO objective over one or more epochs of training with the chosen schedule, and 3) using these observations to update the GP model and select a new $\boldsymbol{\beta}$. Our bandit algorithm is optimal in the sense of converging to a global regret-minimizing solution, as in the time-varying GP bandit optimization approach [5]. By choosing $\boldsymbol{\beta}$ to maximize an acquisition function that balances exploration and performance, our algorithm achieves global guarantees despite the non-convexity of the reward function. Further, our approach is directly aligned with the goal of improved model learning and inference, as the bandit reward function tracks the variational objective over the course of training.

We review the TVO framework in §2, before presenting our bandit optimization approach in §3. We provide details of our time-varying Gaussian process model and discuss its convergence properties in §4. Finally, we demonstrate that our method can improve both model learning and inference in Variational Autoencoders and Sigmoid Belief Networks, in §5.

## 2 The Thermodynamic Variational Objective (TVO)

Assuming a generative model $p_\theta(\mathbf{x}, \mathbf{z})$, we are interested in maximizing the log-likelihood $\log p_\theta(\mathbf{x}) = \log \int p_\theta(\mathbf{x}, \mathbf{z}) d\mathbf{z}$ over parameters $\theta$, given the empirical data $\mathbf{x}$. However, this is intractable due to the integral over the latent variables $\mathbf{z}$. Variational inference methods [4] often seek to maximize the tractable ELBO instead, obtained by introducing an approximate posterior $q_\phi(\mathbf{z} \,|\, \mathbf{x})$ and optimizing the objective

$$\text{ELBO}(\theta, \phi, \mathbf{x}) = \log p_\theta(\mathbf{x}) - D_{\text{KL}}[q_\phi(\mathbf{z} \,|\, \mathbf{x}) \,||\, p_\theta(\mathbf{z} \,|\, \mathbf{x})] = \mathbb{E}_{q_\phi(\mathbf{z} \,|\, \mathbf{x})} \left[ \log \frac{p_\theta(\mathbf{x}, \mathbf{z})}{q_\phi(\mathbf{z} \,|\, \mathbf{x})} \right] . \quad (1)$$

Thermodynamic Integration (TI) [32, 10, 11] is a common technique for estimating (ratios of) partition functions in statistical physics, which instead frames estimating $\log p_\theta(\mathbf{x})$ as a one-dimensional integral over a geometric mixture curve parameterized by $\beta$.[1] In particular, for the TVO, this curve interpolates between the approximate posterior $q_\phi(\mathbf{z} \,|\, \mathbf{x})$ and true posterior $p_\theta(\mathbf{z} \,|\, \mathbf{x})$. Following [6]

this mixture curve can be interpreted as an exponential family of distributions over $\mathbf{z}$ given $\mathbf{x}$

$$\pi_\beta(\mathbf{z} \,|\, \mathbf{x}) = q_\phi(\mathbf{z} \,|\, \mathbf{x}) \exp\{\beta \cdot \log \frac{p_\theta(\mathbf{x}, \mathbf{z})}{q_\phi(\mathbf{z} \,|\, \mathbf{x})} - \log Z_\beta(\mathbf{x})\} \tag{2}$$

$$\text{where } Z_\beta(\mathbf{x}) = \int q_\phi(\mathbf{z} \,|\, \mathbf{x})^{1-\beta} p_\theta(\mathbf{x}, \mathbf{z})^\beta d\mathbf{z} \ .$$

Noting that $\log Z_0(\mathbf{x}) = 0$ and $\log Z_1(\mathbf{x}) = \log p_\theta(\mathbf{x})$, TI now applies the fundamental theorem of calculus to write the model evidence as an integral

$$\log p_\theta(\mathbf{x}) - 0 = \int_0^1 \frac{\partial}{\partial \beta} \log Z_\beta(\mathbf{x}) d\beta = \int_0^1 \mathbb{E}_{\pi_\beta} \left[ \log \frac{p_\theta(\mathbf{x}, \mathbf{z})}{q_\phi(\mathbf{z} \,|\, \mathbf{x})} \right] d\beta, \tag{3}$$

where we have used the known property of exponential families [43] that the derivative of $\log Z_\beta(\mathbf{x})$ with respect to $\beta$ matches the expected sufficient statistics [26, 6]. Masrani et al. [26] use self-normalized importance sampling (SNIS) to estimate each term in the integrand, with $S$ importance samples and $q_\phi(\mathbf{z} \,|\, \mathbf{x})$ as the proposal for each $\beta$

$$\mathbb{E}_{\pi_\beta}[\cdot] \approx \sum_{\ell=1}^{S} \frac{w_\ell^\beta}{\sum_\ell w_\ell^\beta}[\cdot], \quad w_\ell = \frac{p_\theta(\mathbf{x}, \mathbf{z}_\ell)}{q_\phi(\mathbf{z}_\ell \,|\, \mathbf{x})}, \ \mathbf{z}_\ell \sim q_\phi(\mathbf{z} \,|\, \mathbf{x}). \tag{4}$$

Since $\log Z_\beta(\mathbf{x})$ is convex [43], we know that the integrand in (3) is an increasing function of $\beta$. Thus, we can obtain lower and upper bounds using left- and right-Riemann sums, respectively, over a discrete partition $\boldsymbol{\beta}$ of the unit interval. The left-Riemann sum then defines the TVO lower bound

$$\text{TVO}(\theta, \phi, \boldsymbol{\beta}, \mathbf{x}) := \sum_{j=0}^{d-1} (\beta_{j+1} - \beta_j) \, \mathbb{E}_{\pi_{\beta_j}} \left[ \log \frac{p_\theta(\mathbf{x}, \mathbf{z})}{q_\phi(\mathbf{z} \,|\, \mathbf{x})} \right], \tag{5}$$

where $\boldsymbol{\beta} = [\beta_j]_{j=0}^{d-1}$ with $\beta_0 = 0$ and $\beta_j < \beta_{j+1}$. Note that the single-term left-Riemann sum with $\boldsymbol{\beta} = \beta_0 = 0$ matches the ELBO in (1), since $\pi_0(\mathbf{z} \,|\, \mathbf{x}) = q_\phi(\mathbf{z} \,|\, \mathbf{x})$. However, how to choose intermediate $\beta_j$ for $d > 1$ remains an interesting question, which we proceed to frame as a bandit problem.

## 3   From Evidence Maximization to Regret Minimization

We view the vector $\boldsymbol{\beta} \in [0, 1]^d$ as an arm [1] to be pulled in a continuous space, given a fixed resource of $T$ training epochs. After each round, we receive an estimate of the log evidence $\mathcal{L}$, from which we will construct a reward function. An important feature of our problem is that the integrand in Figure 1 changes between rounds as training progresses. Thus, our multi-armed bandit problem is said to be *time-varying*, in that the optimal arm and reward function depend on round $t$.

More formally, we define the time-varying reward function $f_t : [0, 1]^d \to \mathbb{R}$ which takes an input $\boldsymbol{\beta}_t$ and produces reward $f_t(\boldsymbol{\beta}_t)$. At each round we get access to a noisy reward $y_t = f_t(\boldsymbol{\beta}_t) + \epsilon_t$ where we assume Gaussian noise $\epsilon_t \sim \mathcal{N}(0, \sigma_f^2)$. We aim to maximize the cumulative reward $\sum_{t=1}^{T/w} f_t(\boldsymbol{\beta}_t)$ across $T/w$ rounds, where $w$ is a divisor of $T$ and will later control the ratio of bandit rounds $t$ to training epochs $i$.

Maximizing the cumulative reward is equivalent to minimizing the *cumulative regret*

$$R_{T/w} := \sum_{t=1}^{T/w} f_t(\boldsymbol{\beta}_t^*) - f_t(\boldsymbol{\beta}_t), \tag{6}$$

where $r_t := f_t(\boldsymbol{\beta}_t^*) - f_t(\boldsymbol{\beta}_t)$ is the *instantaneous regret* defined by the difference between the received reward $f_t(\boldsymbol{\beta}_t)$ and maximum reward attainable $f_t(\boldsymbol{\beta}_t^*)$ at round $t$. The regret, which is non-negative and monotonic, is more convenient to work with than the cumulative reward and will allow us to derive upper bounds in §4.3.

In order to translate the problem of maximizing the log evidence as a function of $\boldsymbol{\beta}$ into the bandits framework, we define a time-varying reward function $f_t(\boldsymbol{\beta}_t)$. We construct this reward in such a way

---

**Algorithm 1** GP-bandit for TVO (high level)

---

Input: schedule dimension $d$, reward function $f_t(\boldsymbol{\beta}_t)$ where $\boldsymbol{\beta}_t \in [0,1]^d$, update frequency $w$
1: **for** $t = 1....T$ **do**
2:    Train the TVO and get $\mathcal{L}_t$ from Eq. (5) given previously obtained $\boldsymbol{\beta}_{t-1}$
3:    **if** $\mathrm{mod}(t,w) = 0$ : time to update $\boldsymbol{\beta}_t$ **then**
4:       Estimate the utility $y_t = \mathcal{L}_t - \mathcal{L}_{t-w}$ and augment $D_t = D_{t-1} \cup (\boldsymbol{\beta}_{t-1}, t, y_t)$
5:       Fit a time-varying, permutation invariant GP to $D_t$
6:       Estimate $\mu_t(\boldsymbol{\beta}), \sigma_t(\boldsymbol{\beta})$ from Eqs. (15,16)
7:       Select $\boldsymbol{\beta}_t = \arg\max \mu_t(\boldsymbol{\beta}) + \sqrt{\kappa_t}\sigma_t(\boldsymbol{\beta})$ where $\kappa_t$ is from Theorem 1
8:    **end if**
9: **end for**

---

that minimizing the cumulative regret is equivalent to maximizing the final log evidence estimate $\mathcal{L}_T := \log p_{\theta_T}(\mathbf{x})$, i.e., such that $\min R_{T/w} = \max \mathcal{L}_T$.

Such a reward function can be defined by partitioning the $T$ training epochs into windows of equal length $w$, and defining the reward for each window $t \in \{0, 1, ..., T/w - 1\}$

$$f_t(\boldsymbol{\beta}_t) := \mathcal{L}_{w(t+1)} - \mathcal{L}_{wt} \tag{7}$$

as the difference between the TVO log evidence estimate one window-length in the future $\mathcal{L}_{w(t+1)}$ and the present estimate $\mathcal{L}_{wt}$. Then, the cumulative reward is given by a telescoping sum over windows

$$\sum_{t=0}^{T/w - 1} f_t(\boldsymbol{\beta}_t) = (\mathcal{L}_w - \mathcal{L}_0) + (\mathcal{L}_{2w} - \mathcal{L}_w) + ... + (\mathcal{L}_{(T/w)w} - \mathcal{L}_{(T/w - 1)w}) \tag{8}$$

$$= \mathcal{L}_T - \mathcal{L}_0, \tag{9}$$

where $\mathcal{L}_0$ is the initial (i.e. untrained) loss. Recalling the definition of cumulative regret in Eq. (6),

$$\min R_{T/w} = \min \left( \sum_{t=0}^{T/w - 1} f_t(\boldsymbol{\beta}_t^*) \right) - \left( \sum_{t=0}^{T/w - 1} f_t(\boldsymbol{\beta}_t) \right) \tag{10}$$

$$= \min \left( (\mathcal{L}_T^* - \mathcal{L}_0) - (\mathcal{L}_T - \mathcal{L}_0) \right) \tag{11}$$

$$= \min (\mathrm{const} - \mathcal{L}_T) \tag{12}$$

$$= \max \mathcal{L}_T. \tag{13}$$

Therefore minimizing the cumulative regret for the reward function defined by Eq. (7) is equivalent to maximizing the log evidence on the final epoch. Next, we describe how to design an optimal decision mechanism to minimize the cumulative regret $R_{T/w}$ using Gaussian processes.

## 4    Minimizing Regret with Gaussian Processes

There are two unresolved problems with the reward function defined in Eq. (7) which still must be addressed. The first is that it is not in fact computable, due to its use of future observations. The second is that it ignores the ordering constraint required for $\boldsymbol{\beta}$ to be a valid Riemann partition.

We can handle both by problems by using a permutation-invariant Gaussian process to form a surrogate for the reward function $f_t(\boldsymbol{\beta}_t)$. The surrogate model will be updated by past rewards, and used in place of $f_t(\boldsymbol{\beta}_t)$ to select the next schedule at the current round, as described in Algorithm 1.

In §4.1 we formally define how to use (time-varying) Gaussian processes in bandit optimization, before describing how our permutation-invariant kernel can be used to solve the problem of ordering constraints on $\boldsymbol{\beta}$ in §4.2. Finally in §4.3 we provide a theoretical guarantee that our bandit optimization will converge to the regret-minimizing choice of $\boldsymbol{\beta}$.

## 4.1 Time-varying Gaussian processes for Bandit Optimization

A popular design in handling time-varying functions [20, 40, 19, 31] such as $f_t(\boldsymbol{\beta}_t)$ is to jointly model the spatial and temporal dimensions using a product of covariance functions $k = k_\beta \otimes k_T$, where $k_\beta : [0,1]^d \times [0,1]^d \to \mathbb{R}$ is a spatial covariance function over actions, $k_T : \mathbb{N} \times \mathbb{N} \to \mathbb{R}$ is a temporal covariance function, and $k : \mathbb{R}_+^{d+1} \times \mathbb{R}_+^{d+1} \to \mathbb{R}$.

Under this joint modeling framework, the GP is defined as follows. At round $t$ we have the history of rewards $\boldsymbol{y}_t = [y_0, ..., y_t]^T$ and sample points $\boldsymbol{X}_t = \{\boldsymbol{x}_0, ..., \boldsymbol{x}_t\}$, where we define $\boldsymbol{x}_t \in \mathbb{R}^{d+1}$ to be the concatenation of $\boldsymbol{\beta}_t$ and timestep $t$, i.e $\boldsymbol{x}_t := [\boldsymbol{\beta}_t, t]^T$. Then the time-varying reward function is GP-distributed according to

$$f_t \sim GP\big(0, k(\boldsymbol{x}, \boldsymbol{x}')\big) \qquad \text{where } k(\boldsymbol{x}, \boldsymbol{x}') := k_\beta(\boldsymbol{\beta}, \boldsymbol{\beta}') \times k_T(t, t'), \tag{14}$$

where we have assumed zero prior mean for simplicity. For theoretical convenience we follow [5] and choose $k_T(t, t') = (1 - \omega)^{\frac{|t - t'|}{2}}$, where $\omega$ is a "forgetting-remembering" trade-off parameter learned from data. We describe $k_\beta(\boldsymbol{\beta}, \boldsymbol{\beta}')$ in §4.2.

Using standard Gaussian identities [3, 33], the posterior predictive is also GP distributed, with mean and variance given by

$$\mu_t(\boldsymbol{\beta}_*) = \boldsymbol{k}_t(\boldsymbol{\beta}_*)^T \left(\boldsymbol{K}_t + \sigma_f^2 \boldsymbol{I}\right)^{-1} \boldsymbol{y}_t \tag{15}$$

$$\sigma_t^2(\boldsymbol{\beta}_*) = \boldsymbol{k}_t(\boldsymbol{\beta}_*, \boldsymbol{\beta}_*) - \boldsymbol{k}_t(\boldsymbol{\beta}_*)^T \left(\boldsymbol{K}_t + \sigma_f^2 \boldsymbol{I}\right)^{-1} \boldsymbol{k}_t(\boldsymbol{\beta}_*) \tag{16}$$

where $\boldsymbol{k}_t(\boldsymbol{\beta}) = [k(\boldsymbol{x}_0, \boldsymbol{x}), ..., k(\boldsymbol{x}_t, \boldsymbol{x})]^T$ and $\boldsymbol{K}_t = [k(\boldsymbol{x}, \boldsymbol{x}')]_{\boldsymbol{x}, \boldsymbol{x}' \in \boldsymbol{X}_t}$. Using this permutation-invariant, time-varying GP we can select $\boldsymbol{\beta}_{t+1}$ by maximizing a linear combination of the GP posterior mean and variance w.r.t $\boldsymbol{\beta}_t$

$$\boldsymbol{\beta}_{t+1} = \arg\max_{\boldsymbol{\beta}_t} \mu_t(\boldsymbol{\beta}_t) + \sqrt{\kappa_t} \sigma_t(\boldsymbol{\beta}_t), \tag{17}$$

where Eq. (17) is referred to as an *acquisition function* and $\kappa_t$ is its exploration-exploitation trade-off parameter. We note that there are other acquisition functions available [15, 13, 14]. Our acquisition function, Eq. (17), is the time-varying version of GP-UCB [39, 5], which allows us to obtain convergence results in §4.3 and set $\kappa_t$ in Theorem 1.

## 4.2 Ordering Constraints and Permutation Invariance

Recall that the vector $\boldsymbol{\beta} = [\beta_0, ..., \beta_{d-1}]^T$ holds the locations of the left Riemann integral approximation in Eq. (5). In order for the left Riemann approximation to the TVO to be sensible, there must be an ordering constraint imposed on $\boldsymbol{\beta}$ such that $0 < \beta_1 < ... < \beta_{d-1} < 1$. We model this in our GP using a projection operator $\Phi$ which imposes the constraint by sorting the vector $\boldsymbol{\beta}$. Applying $\Phi$ within the spatial kernel, we obtain

$$k_\beta(\boldsymbol{\beta}, \boldsymbol{\beta}') := k_S\big(\Phi(\boldsymbol{\beta}), \Phi(\boldsymbol{\beta}')\big). \tag{18}$$

This projection does not change the value of our acquisition function, and maintains the positive definite for any covariance function for the spatial $k_S$, e.g., Matern, Polynomial. We then optimize the acquisition function via a projected-gradient approach. If a $\boldsymbol{\beta}_t$ iterate leaves the feasible set after taking a gradient step, we project it back into the feasible set using $\Phi$ and continue. We note that existing work in the GP literature has considered such projection operations in various contexts [37, 42].

## 4.3 Convergence Analysis

In Eq. (10), we showed that maximizing the TVO objective function $\mathcal{L}_\mathcal{T}$ as a function of $\boldsymbol{\beta}_t$ is equivalent to minimizing the cumulative regret $R_{T/w}$ by sequential optimization within the bandit framework. Here, the subscript $T/w$ refers to the number of bandit updates given the maximum epochs $T$ and the update frequency $w$ where $w \ll T$.

We now derive an upper bound on the cumulative regret, and show that it asymptotically goes to zero as $T$ increases, i.e., $\lim_{T \to \infty} \frac{R_{T/w}}{T} = 0$. Thus, our bandit will converge to choosing $\boldsymbol{\beta}_T$ which yields the optimal value of the TVO objective $\mathcal{L}_\mathcal{T}^*$ for model parameters at step $T$.

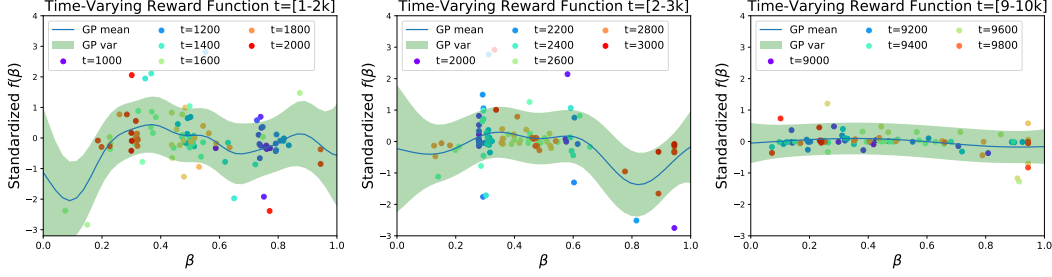

Figure 2: Time-varying reward function $f_t(\boldsymbol{\beta}_t)$ after 2k, 3k, and 10k training epochs, with bandit choices of a single intermediate $\beta_1$ (i.e. $\boldsymbol{\beta} = [0, \beta_1]$) colored by timestep. Scattered $\beta_1$ in neighboring epochs indicate 'exploration', while similarly colored values of $\beta_1$ in regions where the GP mean, or predicted reward, is high indicate 'exploitation'.

We present the main theoretical result in Theorem 1. Our TVO framework mirrors the standard time-varying GP bandit optimization, and thus inherits convergence guarantees from Bogunovic et al. [5]. However, as discussed in Appendix §C, we provide a tighter bound on the mutual information gain $\gamma_{T/w}$ which may be of wider interest.

**Theorem 1.** *Let the domain $\mathcal{D} \subset [0, 1]^d$ be compact and convex. Let $L_t \geq 0$ be the Lipschitz constant for the reward function at time $t$. Assume that the covariance function $k$ is almost surely continuously differentiable, with $f \sim GP(0, k)$. Further, for $t \leq T$ and $j \leq d$, we assume*

$$Pr\left(\sup \left|\partial f_t(\boldsymbol{\beta}_t)/\partial \boldsymbol{\beta}_t^{(j)}\right| \geq L_t\right) \leq ae^{-(L_t/b)^2}$$

*for appropriate choice of $a$ and $b$ corresponding to $L_t$.*

*For $\delta \in (0, 1)$, we write $\kappa_{T/w} = 2\log\frac{\pi^2 T^2}{2\delta w^2} + 2d\log db\frac{T^2}{w^2}\sqrt{\log\frac{da\pi^2 T^2}{2\delta w^2}}$ and $C_1 = 8/\log(1 + \sigma_f^2)$. Then, after $T/w$ time steps, our algorithm satisfies*

$$R_{T/w} = \sum_{t=1}^{T/w} f_t(\boldsymbol{\beta}_t^*) - f_t(\boldsymbol{\beta}_t) \leq \sqrt{\gamma_{T/w} \cdot C_1 \cdot \kappa_{T/w} \cdot T/w} + 2$$

*with probability at least $1 - \delta$, where $\gamma_{T/w}$ is the maximum information gain for the time-varying covariance function (see below).*

In the above theorem, the quantity $\gamma_{T/w}$ measures the maximum information gain obtained about the reward function after pulling $T/w$ arms [39, 5]. In the Appendix §C, we show that $\gamma_{T/w} \leq \left(1 + T/\left\lceil w\tilde{N}\right\rceil\right)\left(\gamma_{\tilde{N}}^{\boldsymbol{\beta}} + \sigma_f^{-2}\tilde{N}^{5/2}\omega\right)$, where $\tilde{N} \in \{1, ..., T/w\}$ denotes a time-varying block length, and $\gamma_{\tilde{N}}^{\boldsymbol{\beta}}$ is defined with respect to the covariance kernel for $\boldsymbol{\beta}$. For our particular choice of exponentiated-quadratic kernel, the maximum information gain scales as $\gamma_{\tilde{N}}^{\boldsymbol{\beta}} \leq \mathcal{O}(\log \tilde{N}^{d+1})$ [39]. Compared with [5], our proof tightens the upper bound on $\gamma_{T/w}$ from $\mathcal{O}(\tilde{N}^3)$ to $\mathcal{O}(\tilde{N}^{5/2})$.

Combining these terms, we can then write the bound as $R_{T/w} \lesssim \mathcal{O}(\sqrt{\left(\left[\log \tilde{N}^{d+1} + \sigma_f^{-2}\tilde{N}^{5/2}\omega\right] T/w\right)})$, which is sublinear in $T$ when the function $f$ becomes time-invariant, i.e., $\omega \to 0$. In contrast, the sublinear guarantee does not hold when the time-varying function is non-correlated, i.e., $\omega = 1$, in which case the time covariance matrix becomes identity matrix. The bound is tighter for lower schedule dimension $d$.

## 5 Experiments

We demonstrate the effectiveness of our method for training VAEs [17] on MNIST and Fashion MNIST, and a Sigmoid Belief Network [27] on binarized MNIST and binarized Omniglot, using the TVO objective. In Appendix D, we explore learning and inference in a discrete probabilistic context-free grammar [23], showing that the TVO objective and our bandit optimization can translate to other

learning settings. In addition, we run ablation studies using random choices of $\boldsymbol{\beta}$ and a GP without permutation invariance, and compare the runtime and performance of our method with grid search. Our code is available at `http://github.com/ntienvu/tvo_gp_bandit`.

**Experimental Setup:** We evaluate our GP-bandit for $S \in \{10, 50\}$ and $d \in \{2, 5, 10, 15\}$ and, for each configuration, train until convergence using 5 random seeds. Note that, for each setting of $d$, we implicitly include $\beta_0 = 0$ and append 1 to the vector $\boldsymbol{\beta}$ to perform the integration in Eq. (5).

For each $S, d$ configuration, we compare against three baseline integration schedules: log-uniform spacing in the interval $[\beta_1, 1]$, linear-uniform spacing in the interval $[0, 1]$, and the moments schedule of [6], which corresponds to uniform spacing along the y-axis. For log/linear-uniform spacing, we set $\beta_1 = 0.025$ for all experiments, reflecting the results of grid search in [26]. We use a fixed model architecture for all experiments, which we describe in Appendix A.

To obtain the bandit feedback in Eq. (7), we use a fixed, linear schedule with $d = 50$ for calculating $\mathcal{L}_t$ with Eq. (5). This yields a tighter $\log p_\theta(\mathbf{x})$ bound, decouples reward function evaluation from model training and schedule selection in each round, and is still efficient using SNIS in Eq. (4). We limit the value of $d$ for TVO training following observations of deteriorating performance in [26].

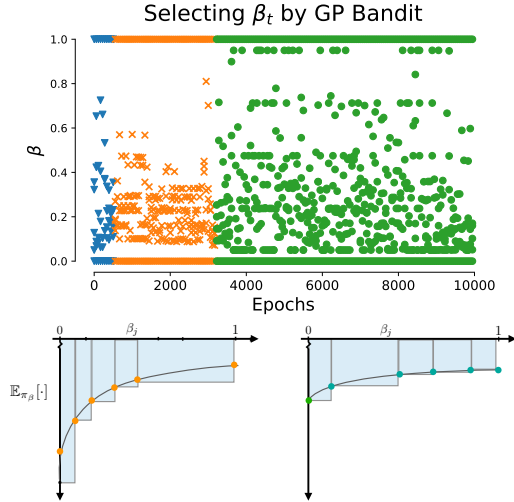

Figure 3: Bandit-chosen $\boldsymbol{\beta}$ over time on MNIST using $d = 5$. We can interpret the $\boldsymbol{\beta}$ selection process in 3 phases: (blue) random selection in initial epochs; (orange) focusing on small values $\beta_i < 0.5$ as training progresses; (green) moving toward $\beta_i = 1$ as learning approaches convergence. The bottom panel illustrates a hypothetical integrand curve and $\boldsymbol{\beta}$ selections at intermediate (left) and later (right) epochs.

**GP Implementation:** For GP modeling, we use an exponentiated quadratic covariance function for $k_\beta$ and estimate hyperparameters via type II maximum likelihood estimation [33]. We use multi-start BFGS [9] to optimize the acquisition function in Eq. (17). We set the update frequency $w = 6$ initially and increment $w$ by one after every 10 bandit iterations to account for smaller objective changes later in training, and update early if $\mathcal{L}_t \leq -0.05$. We found that selecting $\beta_j$ too close to either 0 or 1 could negatively affect performance, and thus restrict $\boldsymbol{\beta} \in [0.05, 0.95]^d$ in all experiments. We follow a common practice to standardize with the running average the utility score $y \sim \mathcal{N}(0, 1)$ for robustness.

## 5.1 Scheduling Behaviour

We first investigate the behaviour of our time-varying reward function and bandit scheduling. These experiments highlight the adaptive nature of our algorithm, as we inspect the choice of integration schedule across training epochs for both $d = 2$ and $d = 5$.

**Time-varying Reward Function:** In Figure 2, we visualize the mean and variance of our time-varying estimate of the utility function $y_t = f_t(\boldsymbol{\beta}_t) + \epsilon_t$ after 2000, 3000, and 10,000 epochs, respectively. We illustrate the choice of $\boldsymbol{\beta}$ for $d = 2$, so that $\beta_0 = 0$ is fixed and we can write the reward as $f_t(\beta_1)$. Colored dots indicate values of $\beta_1$ selected by our bandit algorithm in each round, with the vertical axis reflecting the observed reward $f_t(\beta_1)$ as the change in model evidence $\mathcal{L}$.

In the first two panels, we observe instances where our bandit prioritizes exploitation, choosing similar, high-reward $\beta_1$ values in neighboring rounds with the same color. However, note that these $\beta_1$ may not match the highest GP predictive mean for $f_t(\boldsymbol{\beta})$, since the blue line is shown at the final training epoch in a window. In the final panel, we observe that our time-varying reward function has adapted to have very low variance, since the TVO objective changes only slightly near convergence and the choice of $\beta_1$ has little impact.

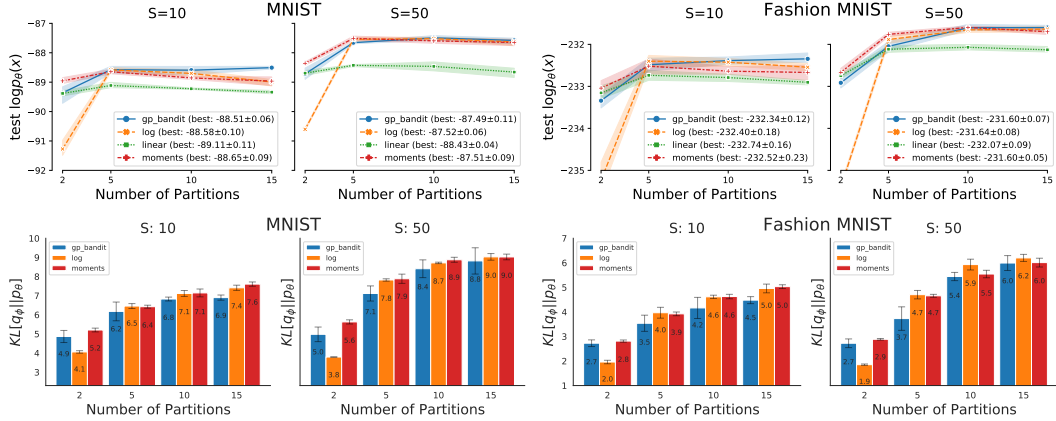

Figure 4: Performance comparison for continuous VAE on MNIST and Fashion MNIST. *Top*: we compare model learning performance using test likelihood (higher is better). *Bottom*: We compare posterior inference as measured by the test KL divergence (lower is better) against the log and moments baselines. Although, in general, we find that models with worse $\log p(\mathbf{x})$ tend to have lower $D_{KL}$, our GP-bandit schedule provides improvements in both learning and inference.

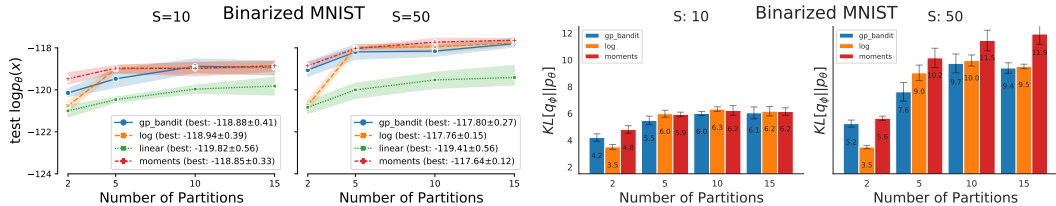

Figure 5: Performance comparison in discrete latent variable model using a Sigmoid Belief Network on the binarized MNIST dataset. Our GP-bandit achieves comparable results to the log and moments schedule in terms model learning (higher $\log p(\mathbf{x})$), with better posterior inference (lower $D_{KL}$).

**$\boldsymbol{\beta}$ Across Training:** In Figure 3, we visualize bandit choices of $\boldsymbol{\beta}$ with $d = 5$. In initial epochs (blue), the GP-bandit algorithm prioritizes exploration before focusing on $\beta_j < 0.5$ in the second phase (orange). As the VAE converges, our algorithm begins to explore $\beta_j$ further from zero (green).

Beyond avoiding the need for an expensive grid search, a primary motivation for our bandit approach is a lack of knowledge about the shape of the integrand. Using the intuition that $\beta_j$ choices should be concentrated in regions where the integrand is changing quickly in order to obtain accurate Riemann approximations, we can still translate the observed bandit choices of $\boldsymbol{\beta}$ into example integrands in the middle (orange) and late (green) stages of training in the bottom panel of Fig. 3. An integrand that rises steeply away from $\beta = 0$ indicates that $q_\phi(\mathbf{z} \,|\, \mathbf{x})$ is mismatched to $p_\theta(\mathbf{z} \,|\, \mathbf{x})$, and the TVO might be improved by choosing small $\beta_j$. As the curve begins to smooth later in training, with a higher proportion of importance samples yielding high likelihood under the generative model, our bandit begins to explore $\beta_j$ closer to 1.

## 5.2 Model Learning and Inference

**Continuous VAE:** We present results of training a continuous VAE on the MNIST and Fashion MNIST dataset in Figure 4. We measure model learning performance using the test log evidence, as estimated by the IWAE bound [7] with 5000 samples per data point. We also compare inference performance using $D_{\mathrm{KL}}[q_\phi(\mathbf{z} \,|\, \mathbf{x}) \,||\, p_\theta(\mathbf{z} \,|\, \mathbf{x})]$, which we calculate by subtracting the test ELBO from our estimate of $\log p_\theta(\mathbf{x})$.

For most scenarios in Figure 4, our GP bandit optimization outperforms baselines with respect to both model learning and inference. In general, we observe that models with lower model evidence attain lower test KL divergence. Thus, in comparing inference performance in the bottom panel of Figure 4, we compare against the log and moment schedules, baselines with comparable test log

likelihoods. It is notable that our approach often achieves better results for both learning (higher $\log p_\theta(\mathbf{x})$) and inference (lower $D_{\text{KL}}$). We obtain the highest log evidence with $d = 10$ for MNIST and $d = 15$ for Fashion MNIST.

**Sigmoid Belief Network:** We present similar results for learning discrete latent variable models using a Sigmoid Belief Network [27]. We show results on binarized MNIST in Figure 5, with binarized Omniglot in Figure 7 in Appendix D.2. Our GP bandit optimization achieves competitive model learning performance with the log-uniform and moment schedules and better posterior inference across models with comparable $\log p(\mathbf{x})$, indicating our GP-bandit schedule can flexibly optimize the TVO for various model types.

## 6 Conclusion

We have presented a new approach for automated selection of the integration schedule for the Thermodynamic Variational Objective. Our bandit framework optimizes a reward function that is directly linked to improvements in the generative model evidence over the course of training the model parameters. We show theoretically that this procedure asymptotically minimizes the regret as a function of the choice of schedule. Finally, we demonstrated that the proposed approach empirically outperforms existing schedules in both model learning and inference for discrete and continuous generative models.

Our GP bandit optimization offers a general solution to choosing the integration schedule in the TVO. However, our algorithm, as well as all other existing schedules, still rely on the number of partitions $d$ as a hyperparameter which is fixed over the course of the training. Incorporating the adaptive selection of $d$ into our bandit optimization remains an interesting direction for future work.

## 7 Broader Impact

Our research can be widely applied for variational inference in deep generative models, including variational autoencoders with autoregressive decoders and normalizing flows. Variational inference, and Bayesian methods more generally, have broad applications spanning science and engineering, from epidemiology [44] to particle physics [2]. Our methodological contributions for variational inference may find broader impact through improved modelling in these disparate domains. However, our method is general in nature, so domain-specific applications should further consider implications for deployment in the real-world.

## 8 Acknowledgements

VM acknowledges the support of the Natural Sciences and Engineering Research Council of Canada (NSERC) under award number PGSD3-535575-2019 and the British Columbia Graduate Scholarship, award number 6768. VM/FW acknowledge the support of the Natural Sciences and Engineering Research Council of Canada (NSERC), the Canada CIFAR AI Chairs Program, and the Intel Parallel Computing Centers program. RB acknowledges support from the Defense Advanced Research Projects Agency (DARPA) under award FA8750-17-C-0106.

This material is based upon work supported by the United States Air Force Research Laboratory (AFRL) under the Defense Advanced Research Projects Agency (DARPA) Data Driven Discovery Models (D3M) program (Contract No. FA8750-19-2-0222) and Learning with Less Labels (LwLL) program (Contract No.FA875019C0515). Additional support was provided by UBC's Composites Research Network (CRN), Data Science Institute (DSI) and Support for Teams to Advance Interdisciplinary Research (STAIR) Grants. This research was enabled in part by technical support and computational resources provided by WestGrid (`https://www.westgrid.ca/`) and Compute Canada (`www.computecanada.ca`).

## Footnotes

[1]Here $\beta$ is a scalar to be consistent with notation in [26] In the remainder of the paper, we let $\boldsymbol{\beta}$ hold the sorted vector of discretization points $\boldsymbol{\beta} = [\beta_0, \beta_1, ..., \beta_{d-1}]^T$, so that each $\beta_j$ specifies a $\pi_{\beta_j}(\mathbf{z} \,|\, \mathbf{x})$ in (2).

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
