[Supplementary Material]

## A  Experimental Setup

**Dataset Description**   The discrete and continuous VAE literature use slightly different training procedures. For continuous VAEs, we follow the sampling procedure described in footnote 2, page 6 of Burda et al. [7], and sample binary-valued pixels with expectation equal to the original gray scale $28 \times 28$ image. We split MNIST [24] and Fashion MNIST [45] into 60k training examples and 10k testing examples across 10 classes.

For Sigmoid Belief Networks, we follow the procedure described by Mnih and Rezende [28] and use 50k training examples and 10k testing examples (the remaining 10k validation examples are not used) with the binarized MNIST [36] dataset.

**Training Procedure**   All models are written in PyTorch and trained on GPUs. For each scheduler, we train for $10,000$ epochs using the Adam optimizer [16] with a learning rate of $10^{-3}$, and minibatch size of 1000. All weights are initialized with PyTorch's default initializer. For the neural network architecture, we use two hidden layers of $[100, 25]$ nodes.

**Reward Evaluation**   To obtain the bandit feedback in Eq. (7), we use a fixed, linear schedule with $d = 50$ for calculating $\mathcal{L}_t$ with Eq. (5). This yields a tighter $\log p_\theta(\mathbf{x})$ bound, decouples reward function evaluation from model training and schedule selection in each round, and is still efficient using SNIS in Eq. (4). We limit the value of $d$ for TVO training following observations of deteriorating performance in [26].

## B  GP kernels and treatment of GP hyperparameters

We present the GP kernels and treatment of GP hyperparameters for the black-box function $f$.

We use the exponentiated quadratic (or squared exponential) covariance function for input hyperparameter $k_\beta(\boldsymbol{\beta}, \boldsymbol{\beta}') = \exp\left(-\frac{||\boldsymbol{\beta}-\boldsymbol{\beta}'||^2}{2\sigma_\beta^2}\right)$ and a time kernel $k_T(t, t') = (1-\omega)^{\frac{|t-t'|}{2}}$ where the observation $\boldsymbol{\beta}$ and $t$ are normalized to $[0,1]^d$ and the outcome $y$ is standardized $y \sim \mathcal{N}(0,1)$ for robustness. As a result, our product kernel becomes

$$k\left([\boldsymbol{\beta}, t], [\boldsymbol{\beta}', t']\right) = k(\boldsymbol{\beta}, \boldsymbol{\beta}') \times k(t, t') = \exp\left(-\frac{||\boldsymbol{\beta}-\boldsymbol{\beta}'||^2}{2\sigma_\beta^2}\right)(1-\omega)^{\frac{|t-t'|}{2}}.$$

The length-scales $\sigma_\beta$ is estimated from the data indicating the variability of the function with regards to the hyperparameter input $\mathbf{x}$ and number of training iterations $t$. Estimating appropriate values for them is critical as this represents the GP's prior regarding the sensitivity of performance w.r.t. changes in the number of training iterations and hyperparameters. We note that previous works have also utilized the above product of spatial and temporal covariance functions for different settings [20, 5, 29].

We fit the GP hyperparameters by maximizing their posterior probability (MAP), $p(\sigma_l, \omega \mid \boldsymbol{\beta}, \mathbf{t}, \mathbf{y}) \propto p(\sigma_l, \omega, \boldsymbol{\beta}, \mathbf{t}, \mathbf{y})$, which, thanks to the Gaussian likelihood, is available in closed form as [33]

$$\ln p(\mathbf{y}, \boldsymbol{\beta}, \mathbf{t}, \sigma_l, \omega) = \frac{1}{2}\mathbf{y}^T\left(\boldsymbol{K} + \sigma^2\mathbf{I}_N\right)^{-1}\mathbf{y} - \frac{1}{2}\ln\left|\boldsymbol{K} + \sigma^2\mathbf{I}_N\right| + \ln p_{\text{hyp}}(\sigma_x, \omega) + \text{const} \quad (19)$$

where $\mathbf{I}_N$ is the identity matrix in dimension $N$ (the number of points in the training set), and $p_{\text{hyp}}(\sigma_l, \omega)$ is the prior over hyperparameters, described in the following.

We maximize the marginal likelihood in Eq. (19) to select the suitable lengthscale parameter $\sigma_l$, remembering-forgetting trade-off $\omega$, and noise variance $\sigma_f^2$.

Optimizing Eq. (19) involves taking the derivative w.r.t. each variable, such as $\frac{\partial \ln p(\mathbf{y}, \boldsymbol{\beta}, \mathbf{t}, \sigma_l, \omega)}{\partial \omega} = \frac{\partial \ln p(\mathbf{y}, \boldsymbol{\beta}, \mathbf{t}, \sigma_l, \omega)}{\partial \boldsymbol{K}} \times \frac{\partial \boldsymbol{K}}{\partial k(t, t')} \times \frac{\partial k(t, t')}{\partial \omega}$. While the derivatives of $\sigma_l$ and $\sigma_f^2$ are standard and can be found in [33], we present the derivative w.r.t. $\omega$ as follows

$$\frac{\partial k(t, t')}{\partial \omega} = -v(1-\omega)^{v-1} \text{ where } v = |t-t'|/2. \quad (20)$$

We optimize Eq. (19) with a gradient-based optimizer, providing the analytical gradient to the algorithm. We start the optimization from the previous hyperparameter values $\theta_{prev}$. If the optimization fails due to numerical issues, we keep the previous value of the hyperparameters.

## C   Proof of Theorem 1

Our use of the TVGP within the TVO setting requires no problem specific modifications compared to the general formulation in Bogunovic. As such, the proof of Theorem 1 closely follows the proof of Theorem 4.3 in Bogunovic et al. [5] App. C. with time kernel $k_{\mathrm{T}}(i,j) = (1-\omega)^{\frac{|i-j|}{2}}$. At a high level, their proof proceeds by partitioning the $T$ random functions into blocks of length $\tilde{N}$, and bounding each using Mirsky's theorem. Referring to Table 1 for notation, this results in a bound on the maximum mutual information

$$\tilde{\gamma}_{\tilde{N}} \leq \left(\frac{T}{\tilde{N}} + 1\right)\left(\gamma_{\tilde{N}} + \tilde{N}^3\omega\right),\tag{21}$$

which leads directly to their bound on the cumulative regret (cf. App C.2 in [5]). Our contribution is to recognize we can achieve a tighter bound on the maximum mutual information with an application Cauchy Schwarz and Jensen's inequality.

*Proof.* Beginning from Bogunovic et al. [5] Eq. (58), we have

$$\tilde{\gamma}_{\tilde{N}} \leq \gamma_{\tilde{N}} + \sum_{i=1}^{\tilde{N}} \log\left(1 + \Delta_i\right)\tag{22}$$

$$\tilde{\gamma}_{\tilde{N}} \leq \gamma_{\tilde{N}} + \tilde{N}\log\left(1 + \frac{1}{\tilde{N}}\sum_{i=1}^{\tilde{N}}\Delta_i\right) \qquad \text{Jensen's inequality}\tag{23}$$

$$\tilde{\gamma}_{\tilde{N}} \leq \gamma_{\tilde{N}} + \tilde{N}\log\left(1 + \frac{1}{\sqrt{\tilde{N}}}\sqrt{\sum_{i=1}^{\tilde{N}}\Delta_i^2}\right) \qquad \text{Cauchy-Schwartz}\tag{24}$$

$$\tilde{\gamma}_{\tilde{N}} \leq \gamma_{\tilde{N}} + \tilde{N}\log\left(1 + \tilde{N}^{3/2}\omega\right) \qquad \sum_{i=1}^{\tilde{N}}\Delta_i^2 \leq \tilde{N}^4\omega^2\tag{25}$$

$$\tilde{\gamma}_{\tilde{N}} \leq \gamma_{\tilde{N}} + \tilde{N}^{5/2}\omega \qquad \log(1+x) \leq x\tag{26}$$

This bound is tighter than [5] Eq. (60) $(\tilde{N}^{5/2} \leq \tilde{N}^3)$, where the latter was achieved via a simple constrained optimization argument. Using (26), Theorem 1 follows using identical arguments as in [5]. $\qquad\qquad\square$

## D   Additional Experiments and Ablation Studies

We present additional experiments on a Probabilistic Context Free Grammar (PCFG) model and Sigmoid Belief Networks in §D.1 and §D.2, a wall-clock time benchmark in §D.3, ablation studies in §D.4, and additional training curves in §D.5.

### D.1   Training Probabilistic Context Free Grammar

In order to evaluate our method outside of the Variational Autoencoder framework, we consider model learning and amortized inference in the probabilistic context-free grammar setting described in Section 4.1 of Le et al. [23]. Here $p_\theta(\mathbf{x}, \mathbf{z}) = p(\mathbf{x}\,|\,\mathbf{z})p_\theta(\mathbf{z})$, where $p_\theta(\mathbf{z})$ is a prior over parse trees $\mathbf{z}$, $p(\mathbf{x}\,|\,\mathbf{z})$ is a soft relaxation of the $\{0,1\}$ likelihood which indicates if sentence $\mathbf{x}$ matches the set of terminals (i.e the sentence) produced by $\mathbf{z}$, and $\theta$ is the set of probabilities associated with each production rule in the grammar. The inference network $\phi$ is a recurrent neural network which outputs $q_\phi(\mathbf{z}\,|\,\mathbf{x})$, the conditional distribution of a parse tree given an input sentence. We use

Table 1: Supporting notations in regret analysis. We use notation to similar to Appendix C of Bogunovic et al. [5] when possible.

| Parameter | Domain | Meaning |
|---|---|---|
| $\omega$ | scalar, $(0, 1)$ | Remembering-forgetting trade-off parameter ($\epsilon$ in [5]) |
| $\mathbf{f}_T$ | vector, $\mathbb{R}^T$ | Vector of $T$ function evaluations from $f$, $\mathbf{f_T} := [f(x_1), ..., f(x_T)]^T$. |
| $\tilde{\mathbf{f}}_T$ | vector, $\mathbb{R}^T$ | (*time-varying case*) Vector of $T$ function evaluations from $f_{1:T}$, $\tilde{\mathbf{f}}_T := [f_1(x_1), ..., f_T(x_T)]^T$ |
| $I(\mathbf{y}_T; \mathbf{f}_T)$ | scalar, $\mathbb{R}^+$ | The mutual information in $\mathbf{f}_T$ after revealing $\mathbf{y}_T = \mathbf{f}_T + \epsilon$. For a GP with covariance function $\mathbf{K}_T$, $I(\mathbf{y}_T; \mathbf{f}_T) = \frac{1}{2}\log|\mathbf{I} + \sigma^{-2}\mathbf{K}_T|$ |
| $\tilde{I}(\mathbf{y}_T; \tilde{\mathbf{f}}_T)$ | scalar, $\mathbb{R}^+$ | (*time-varying case*) Mutual information $\tilde{I}(\mathbf{y}_T; \tilde{\mathbf{f}}_T) = \frac{1}{2}\log|\mathbf{I} + \sigma^{-2}\tilde{\mathbf{K}}_T|$, where $\tilde{\mathbf{K}}_T$ is a covariance function that incorporates time kernel $k_T(i, j)$ |
| $\gamma_T$ | scalar, $\mathbb{R}^+$ | The maximum information gain $\gamma_T := \max_{x_1,...,x_T} I(\mathbf{y}_T; \mathbf{f}_T)$ after $T$ rounds |
| $\tilde{\gamma}_T$ | scalar, $\mathbb{R}^+$ | (*time-varying case*) The analogous time-varying maximum information gain $\tilde{\gamma}_T := \max_{x_1,...,x_T} \tilde{I}(\mathbf{y}_T; \tilde{\mathbf{f}}_T)$ |
| $\tilde{N}$ | scalar, $\mathbb{R}^+$ | An artifact of the proof technique used by [5]. The $T$ time steps are partitioned into blocks of length $\tilde{N}$ |

the *Astronomers* PCFG considered by Le et al. [23], and therefore have access to the ground truth production probabilities $\theta_{\text{true}}$, which we will use to evaluate the quality of our learned model $\theta$.

We compare the TVO with GP-bandit and log schedules against REINFORCE, WAKE-WAKE, and WAKE-SLEEP, where WAKE-WAKE and WAKE-SLEEP use data from the true model $\mathbf{x} \sim p_{\theta_{\text{true}}}(\mathbf{x})$ and learned model $\mathbf{x} \sim p_\theta(\mathbf{x})$ respectively. For each run, we use a batch size of 2 and train for 2000 epochs with Adam using default parameters. For all KL divergences (see caption in Figure 6), we compute the median over 20 seeds and then plot the average over the last 100 epochs.

As observed by Le et al. [23], sleep-$\phi$ updates avoid the deleterious effects of the SNIS bias and is therefore preferable to wake-$\phi$ updates in this context. Therefore for all runs we use the TVO to update $\theta$, and use sleep-$\phi$ to update $\phi$. Sleep-$\phi$ is a special case of the TVO (cf. Masrani et al. [26] Appendix G.1).

In Figure 6 we see that both GP-bandits and log schedules have comparable performance in this setting, with TVO-log, $S = 20$ achieving the lowest $\text{KL}[p_\theta || p_{\theta_{\text{true}}}]$ and $\text{KL}[q_\phi(\mathbf{z} \mid \mathbf{x}) || p_{\theta_{\text{true}}}(\mathbf{z} \mid \mathbf{x})]$ across all trials. $\text{KL}[q_\phi || p_{\theta_{\text{true}}}]$ is a preferable metric to $\text{KL}[q_\phi || p_\theta]$ because the former does not depend on the quality of the learned model. We also note that GP-bandits appears to be less sensitive to the number of partitions than the log schedule.

## D.2 Training Sigmoid Belief Network on Binarized Omniglot

We train the Sigmoid Belief Network described in §5.2 on the binarized Omniglot dataset. Omniglot [21] has 1623 handwritten characters across 50 alphabets. We manually binarize the omniglot[22] dataset by sampling once according to procedure described in [7], and split the binarized omniglot dataset into $23,000$ training and $8,070$ test examples. Results are shown in Figure 7. At $S = 50$, GP-bandit achieves similar model learning but better inference compared to log scheduling.

## D.3 Wall-clock time Comparison

We benchmark the wall-clock time of our GP-bandit schedule against the cumulative wall-clock time of the grid-search log schedule. For both schedules we train a VAE on the Omniglot dataset for $S = 10$ and 5000 epochs. For the log schedule, we run the sweep ran by Masrani et al. [26] (cf. section 7.2), i.e. 20 $\beta_1$ linearly spaced between $[10^{-2}, 0.9]$ for $d = 2, ..., 6$, for a total of 100 runs. For a fair comparison against the log schedule, we loop over $d = 2, ..., 6$ for our GP bandits because $d$ is unlearned, for a total of five runs. We note that each run of the GP bandits schedule

Figure 6: Evaluation of model learning in a PCFG, where $\theta$ is the set of probabilities associated with each production rule in the grammar, and $\phi$ is an RNN which generates the conditional probability of a parse tree given a sentence. GP-bandits (ours) is comparable to the baseline log schedule and less sensitive to number of partitions, as evaluated by the KL divergence between learned and true model parameters (top row). Inference network learning is evaluated by the KL divergence between $q_\phi(\mathbf{z} \mid \mathbf{x})$ and $p_{\theta_{\text{true}}}(\mathbf{z} \mid \mathbf{x})$ (middle row) and $p_\theta(\mathbf{z} \mid \mathbf{x})$ (bottom row). We compare against REINFORCE, WAKE-WAKE, and WAKE-SLEEP, where some baselines aren't shown due to being out of range. At $S = 20$, TVO with log and GP-bandits schedule outperforms REINFORCE, WAKE-WAKE, and WAKE-SLEEP both in terms of the quality of the generative model (top row, right) and inference network (middle row, right) for all $S \in \{2, 5, 10, 20\}$.

Figure 7: Performance of the Sigmoid Belief Network described in §5.2 on the binarized omniglot dataset. The GP-bandit schedule at $d = 15$, $S = 50$ outperforms all baselines in terms of model and inference network learning.

includes learning the GP hyperparameters as described in Appendix B. For both schedules, we take the best $\log p(x)$ and corresponding KL divergence, and plot the cumulative run time across all runs. The results in Table 2 show that the GP-bandits schedule does comparable to the grid searched log schedule (log likelihood: $-110.72$ vs $-110.99$) while requiring significantly less cumulative wall-clock time (10 hrs vs 178 hrs).

Figure 8: We compare the performance between our GP-bandit against the Random search (Rand) baseline which uniformly generates the integration schedules $\beta_t$. The GP-bandit schedule outperforms the random counterpart by using information obtained from previous choices, as described in Algorithm 1.

Table 2: Wallclock time of the GP-bandit schedule compared to the grid-search of [26] for the log schedule. GP-bandit approach achieves a competitive test log likelihood and lower KL divergence compared with the grid-searched log schedule, but requires significantly lower cumulative run-time.

|  | best $\log p(x)$ | best kl | number of runs | cumulative run time (hrs) |
|---|---|---|---|---|
| GP bandit (ours) | -110.995 | **7.655** | 5 | **10.99** |
| grid-searched log | **-110.722** | 8.389 | 100 | 177.01 |

## D.4  Ablation studies

**Ablation study between GP-bandit and random search.**  To demonstrate that our model can leverage useful information from past data, we compare against the Random Search picks the integration schedule uniformly at random.

We present the results in Figure 8 using MNIST (left) and Fashion MNIST (right). We observe that our GP-bandit clearly outperforms the Random baseline. The gap is generally increasing with larger dimension $d$, e.g., $d = 15$ as the search space grows exponentially with the dimension.

**Ablation study between permutation invariant GPs.**  We compare our GP-bandit model using two versions of (1) non-permutation invariant GP and (2) permutation invariant GP in Table 3.

Our permutation invariant GP does not need to add all permuted observations into the model, but is still capable of generalizing. The result in Table 3 confirms that if we have more samples to learn the GP, such as using larger epochs budget $T$, the two versions will result in the same performance. On the other hand, if we have limited number of training budgets, e.g., using lower number of epochs, the permutation invariant GP will be more favorable and outperforms the non-permutation invariant. In addition, the result suggests that for higher dimension $d = 15$ (number of partitions) our permutation invariant GP performs consistently better than the counterpart.

## D.5  Training Curves

We show example training curves for $S = 10, d \in \{5, 15\}$ obtained using the linear, log, moment, and GP-bandit schedules in Figure 9. We can see sudden drops in the GP-bandit training curves indicating our model is exploring alternate schedules during training (cf. Section 4.1).

Table 3: Comparison between permutation invariant and non-permutation invariant in MNIST dataset using S=10 (top) and S=50 (bottom). The best scores are in bold. Given $T$ used epochs, the number of bandit update and thus the number of sample for GP is $T/w$ where $w = 10$ is the frequency update. The permutation invariant will be more favorable when we have less samples for fitting the GP, as indicated in less number of used epochs $T = 1000, 2000$. The performance is comparable when we collect sufficiently large number of samples, e.g., when $T/w = 1000$.

| S=10 | Used Epoch $T$/ Bandit Iteration | 1000/100 | 2000/200 | 5000/500 | 10000/1000 |
|---|---|---|---|---|---|
| d=5 | Perm Invariant | **−91.488** | **−90.129** | **−89.130** | −88.651 |
| | Non Perm Invariant | −91.554 | −90.206 | −89.262 | **−88.552** |
| d=10 | Perm Invariant | **−91.430** | **−90.219** | −89.159 | −88.603 |
| | Non Perm Invariant | −91.553 | −90.249 | **−89.110** | **−88.466** |
| d=15 | Perm Invariant | **−91.386** | **−90.059** | **−88.957** | **−88.504** |
| | Non Perm Invariant | −91.550 | −90.224 | −89.215 | −88.564 |
| S=50 | Used Epoch $T$/ Bandit Iteration | 1000/100 | 2000/200 | 5000/500 | 10000/1000 |
| d=5 | Permutation Invariant | **−90.071** | **−89.068** | **−88.163** | −87.979 |
| | Non Permutation Invariant | −90.119 | −89.142 | −88.215 | **−87.860** |
| d=10 | Perm Invariant | **−90.125** | **−89.115** | **−88.187** | −87.859 |
| | Non Permutation Invariant | −90.212 | −89.225 | −88.231 | **−87.702** |
| d=15 | Permutation Invariant | **−90.029** | **−89.082** | **−88.102** | **−87.579** |
| | Non Permutation Invariant | −90.157 | −89.247 | −88.173 | −87.631 |

Figure 9: We plot $\log p(\mathbf{x})$ on the training set throughout $S = 10$, $d \in \{5, 15\}$ for each dataset using the experimental setup described in Appendix A. The final training log likelihoods are consistent with the test log likelihoods. Small drops in the GP-bandit training curves indicate the algorithm exploring the reward landscape (see Section 4.1).