[Reviews · NeurIPS 2020]

Review 1

Summary and Contributions: The authors present a bandit-based method for selecting an integration schedule for the thermodynamic variational objective. They provide theoretical guarantees for their approach and evaluate it empirically on several tasks.

Strengths: This is a lovely paper. It is clear and a pleasure to read. The approach is well thought out and well presented. It is novel and relevant to the NeurIPS community.

Weaknesses: The main weakness of the paper is the strength of the results. The method appears to work, but only offers gains in some settings, and many of those gains are relatively small. Additionally, we do not know if the modest gains in log likelihoods translate to concretely improved samples. It would be nice to see if the method actually results in a model that produces better samples. In addition to this weakness there are several other smaller issues: * The first sentence in the abstract is awkward, feels too parenthetical * The notation is a bit confusing, I would recommend sticking to using expectations with brackets. For example equation 5 has no brackets and equation 4 has some things in brackets and some things outside brackets. * Figure 5 caption seem to have swapped model inference and learning. * I think it would be instructive to see training loss. While the main thrust of the authors' argument is that their method improves model learning, the method's effect on the optimization of the training objective is also relevant. * I would have liked to see more datasets such as CelebA. * Just to satisfy my own curiosity: Instead of sorting the outputs of the Gaussian process and doing projected gradient descent, why not use the outputs of the gaussian process as the width of the integration intervals?

Correctness: I have not verified in-depth the proofs in their appendix, but the main text is correct.

Clarity: Yes, very much so.

Relation to Prior Work: Yes

Reproducibility: Yes

Additional Feedback: POST REBUTTAL UPDATE: Thank you authors for your rebuttal. I did not have significant issues with the paper before rebuttal, and few changes were proposed to address the issues I did have so I will leave my score as-is.


Review 2

Summary and Contributions: Choosing the discrete points for the Riemannian sum is critical for the performance of the recently proposed thermodynamic variational objective. Instead of using linear or log spacing for these points, the paper proposes using bandit to adaptively explore/exploit these values. As the objective changes through time (due to change of the integrand), the paper considers modelling the deltas through time via a time-varying Gaussian process, with an operator to make sure the order of the beta values does not change. The paper provides some theoretical analysis and experiments on generative models with continuous and discrete latent variables.

Strengths: The paper considers an important problem with the recently proposed thermodynamic variational objective, which might be of interest to the Bayesian ML community, as improving the bound estimate might translate to better learning and inference for generative models. The application of GP bandit optimisation is interesting and non-trivial. The use of time-varying Gaussian processes for bandit might be applicable to other settings where correlation through time is important and can be captured via a kernel. The intuition about how beta values should be adaptive is useful to understand more the behaviour of TVO.

Weaknesses: Significance: whilst the application of bandits to this selection problem is interesting, it might be hard for practitioners to adopt this: (i) there are many hyperparameters, notably the update cadence of the bandit, and (ii) the lower bound might be tighter (as demonstrated by the smaller gap between the estimated log likelihood and the bounds, the KL divergence), it does not translate to better predictive performance (except for D = 2). The log-spacing scheme is very simple and seems to perform well in practice. Technical argument: Whilst the integrand is an increasing function of beta, it is hard to know (i) how the shape of this integrand looks like for a fixed variational approximation with different beta values (many steep/shallow slopes or very smooth?), and (ii) how the objective changes from one step to the next step, i.e. how similar/correlated the objectives or the change in the objectives across different optimisation steps and how useful is the decision for beta now towards the future as the integrand might have changed drastically. It would be great if these aspects could be investigated empirically to justify the proposal. It is hard to understand/interpret figures 2 and 3 or use these to make any claim about exploration/exploitation trade-off of the algorithm, or the progression of the selections for beta. Update 1: my concerns have been addressed during the review process. I encourage the authors to improve the clarity of the proposed method and figures to provide more intuition.

Correctness: The theoretical analysis seems correct to me. The method is evaluated on relevant benchmarks.

Clarity: Yes, the paper is nicely structured and written.

Relation to Prior Work: Yes, the new proposed search scheme is positioned well with related literature (linear and log spacing schemes).

Reproducibility: Yes

Additional Feedback:


Review 3

Summary and Contributions: This paper improves the Thermodynamic Variational Objective (TVO) by automatically inferring the integration partition (Beta parameter). Picking the Beta parameter is performed dynamically using Gaussian Process (GP) within a Bandit Optimization framework. The authors provide theoretical guarantees of convergence and show its empirical use.

Strengths: The Thermodynamic Variational Objective is the state-of-the-art gradient estimator for discrete latent variable models, which are fundamental in machine learning. This work solves the main drawback of the TVO: finding the optimal integration partition. Furthermore, the method is interesting in itself as it could be applied more broadly.

Weaknesses: The empirical results show that GPBO+TVO does not always outperform the TVO, which relies on a manually chosen Beta partition. he GPBO method is able to dynamically tune the partition during training, which could theoretically improve learning further. [update] Thank you for your feedback, the use of multiple random seeds (figure 1) shows that GPBO+TVO consistently identifies an effective partition scheme (GPBO+TVO \geq TVO + grid search ), this answers my main concern.

Correctness: I am not in a position that allows me to check the derivations formally. Nonetheless, based on a careful reading of the proofs, I am inclined to judge this work as correct.

Clarity: This work conveys a clear idea, the paper was easy to read despite the complexity of the solution. The experiments are relevant and support the theory.

Relation to Prior Work: Related work is considered sufficiently.

Reproducibility: Yes

Additional Feedback: Questions: how were the uniform partitions defined for each model and number of particles? Are you using the same architecture as in the original TVO paper? Figure 5: Better model inference is translated into a lower KL, and a better model translates into a higher log p(x), not the opposite (the main text description is correct).


Review 4

Summary and Contributions: The key problem addressed is this work is to automate the selection of integration points/schedules for numerically approximating the TVO. The authors undertake a regret minimization approach and use (time-varying) GP-bandit optimization. Theoretical results are provided in addition to experiments on the MNIST, FashionMNIST and Omniglot datasets. == Post-Rebuttal == I have read through the other reviews and the authors' rebuttal. The rebuttal has partially addressed my concerns. I agree with other reviewers that the predictive performance improvement appears minimal. As such, inclusion of metrics (e.g., wall-clock time) to support the claim of efficiency would make clear the paper's contributions That said, my concern about moving the burden to a search over GP hyperparameters remains (R2 appears to share this concern). The log-scheme is indeed simple and already works well. a really fair comparison would include that search in the wall-clock time or at least show insensitivity to a range of hyperparameter choices for the kernels mentioned (exponential quadratic or Matern). I hope the authors can include these in the paper or in the appendix. Overall, I still believe the paper makes a solid enough contribution and my score remains.

Strengths: Overall, the paper presents a solid solution to a known problem when using TVO. The problem formulation is intuitive and coherent, and the solution technique appears to work well compared to a linear schedule. The paper is well-written and presents the key ideas and results in a logical structured manner (up to the points below).

Weaknesses: My concerns are regarding the technical novelty and significance of the work. While the idea of applying bandit optimization to TVO is new, the techniques applied are known and there appears to be little technical innovation in terms of the general methodology. Can the authors better clarify the differences to existing work, e.g. [5]? If there are none, then the contribution is primarily on improving TVO, which while interesting, does limit the significance of the work. On a related note, the method shifts the burden of specifying a fixed schedule to defining an appropriate GP kernel (and potentially, hyperparameters). Of course, the hope is that the GP-bandit will be more broadly applicable. Can the authors clarify the sensitivity of the method to kernel choice and other design considerations? How does the sensitivity compare to using say, a fixed log-schedule (without a large grid search, or with a small set of candidate schedules)? Although the method is well-described, the experiments section could be better explained. What is the key takeaway from Fig 2? The proposed method outperforms a linear schedule, but it does not appear to significantly improve upon a log-uniform baseline for learning. Can the authors indicate how expensive the grid search was relative to the GP-bandit in this setting? Some illustrative comparisons between computational load may help support the use of GP-bandit. Also, were the results obtained from a single run or several? If a single run, what was the variation between different initializations; I’m assuming that data was sampled to compute the TVO objective?

Correctness: I did not find major errors in the paper. However, some issues should be clarified: The covariance between two integration schedules is captured by first projecting them to their correctly ordered counterparts and then capturing the correlation between the projected values using a spatial covariance function. Can any covariance function be used? Can we guarantee that the covariance calculated this way will be positive semi-definite? Section 4 should give some indication on what family of functions to use for spatial covariance. In the main theorem, on the RHS of the bound given on line 171, is \kappa_T correct? Or should it be \kappa_{T/w}? Should the bound hold with 1-\delta or 1-\delta/3? It doesn’t appear if a new \delta was defined and used. What is ref [43] in the supplementary?

Clarity: Yes, the paper is well-written overall.

Relation to Prior Work: The relation to prior work is discussed in text. If space permits, the author should include works on Bayesian Optimization/Bandit Optimization and discuss differences to existing approaches.

Reproducibility: Yes

Additional Feedback:

[Author Response · NeurIPS 2020]

We thank the reviewers for their insightful comments. We first respond to the common concerns across the four reviews before addressing specific reviewer feedback.

**Strength of results, hyperparameters, additional experiments:** All reviewers observe we achieve only modest gains compared to the log-uniform spacing in terms of final log likelihood. We emphasize that while this is true, the main benefit of our approach is in avoiding the costly grid required by the log/linear-uniform spacing schedule. To respond to R4's request, as measured in wall-clock time, our GP-bandit schedule takes approx. 12 hrs to train a VAE on MNIST compared to the approx 160hrs training time for the grid searched log schedule (8 hrs/run x 20 runs). We will update the main text to make these considerations clear, and add an additional appendix section benchmarking our schedule against baselines in terms of total wall-clock time.

To respond to R4, all results were obtained from a single seed. We will update the figures in the final draft to average results across five seeds, and have included preliminary results on MNIST in Figure 1. As per R1's request, we will also include the training loss in the final version, which closely mirrors the test loss curves.

R2 raises a concern about the number of hyper-parameters in our method. The GP-bandit intro-duces three additional hyperparameters which can be learned directly from data by maximizing the GP marginal likelihood, and therefore require no additional hand-tuning by practitioners as discussed in Appendix B. The exploration-exploitation trade off parameter $\kappa_t$ is set using Theorem 1.

Figure 1: VAE MNIST, 6k epochs, 5 seeds

**Figure 2 clarity:** We agree with R2 and R4 that a more thorough description of both Fig. 2 and Fig. 3 are needed. In Fig. 2, we investigate the bandit exploration / exploitation behavior at early, middle, and late stages of training by showing where the bandit positions a single $\beta_1$. Color encodes timestep, so clusters of similar colors indicate the bandit is "exploiting" a particular region, particularly in regions where the blue line, i.e. GP mean, is high. This indicates that our reward model expects the TVO objective will improve with this choice of $\beta_1$. We also show the variance of our GP surrogate model across training phases and see both the mean and variance of predicted reward decreases as optimization converges.

The key takeaway from this figure is that we see the bandit exploits early on in training, and that the surrogate reward function correctly learns that the reward for choosing the correct location decreases as optimization proceeds and the curve flattens.

**Figure 3 clarity:** R2 correctly observes that it is difficult to know the shape of the integrand. We agree, and this is in fact one of our primary motivations for using a bandits schedule, as bandit optimization is uniquely suited to scenarios where one has little knowledge of the form of the reward function. We show possible example shapes for the integrand in the subpanel of Fig. 3, reflecting bandit choices of $\beta$ in the middle (orange) and late (green) stages of training. These are based on the intuition that $\beta$ choices should be concentrated in regions where the integrand is changing quickly, allowing the left Riemann approximation to capture the most area with a fixed budget of $d$ partitions. We also assume that a perfectly flat curve will result in uniform $\beta$ choices. We will update the caption of Fig. 3 to make this clear.

**(R4) Comparison with Bogunovic et. al [5]:** We agree that the positioning of our contribution with respect to [5] requires further clarification, and regret this oversight. Theorem 1 improves the bound on the maximum mutual information gain $\tilde{\gamma}$ from [5] by using Cauchy Schwarz and Jensen's inequality rather than an analysis of the optimality conditions (cf. eq. 61 in [5]). We have updated the main text in 4.3 to make it clear that Theorem 1 follows by using our tighter bound on $\tilde{\gamma}$ in Theorem 4 of [5], and simplified the derivation in the appendix to refer to [5] directly where possible.

**(R4) Discussion on spatial covariance:** R4 asks for clarification on covariance functions which can be used alongside our projection operation. Since the projection preserves the input space, any PSD covariance function will maintain this property after projection. We recommend choosing a kernel for which bounds on the maximum information gain are known, such as exponentiated quadratic, Mattern, or linear from Srinivas et al 2010 [35].

**Typos:** R1, R3, R4 helpfully point out a number of typos and suggestions to improve the writing which we will incorporate into the final draft.

**(R4) Ref [43] in the supplement:** Martin J Wainwright. "Basic concentration bounds." In *High-dimensional Statistics: A non-asymptotic viewpoint.* Chapter 2, pages 21–57. 2019

[Meta-Review · NeurIPS 2020]

Reviewers are positive about the contributions of this paper and feel that the authors addressed their primary concerns. Please update the paper with the revisions proposed in the rebuttal. Further, R4 suggests that a fair comparison would include the search time for the GP hyperparameters or show insensitivity. I agree and think this would strengthen the paper.